# Effects of Contaminated Soil on the Survival and Growth Performance of European (*Populus tremula* L.) and Hybrid Aspen (*Populus tremula* L. *× Populus tremuloides* Michx.) Clones Based on Stand Density

**DOI:** 10.3390/plants11151970

**Published:** 2022-07-28

**Authors:** Mir Md Abdus Salam, Wen Ruhui, Aki Sinkkonen, Ari Pappinen, Pertti Pulkkinen

**Affiliations:** 1School of Forest Sciences, University of Eastern Finland, 80100 Joensuu, Finland; ari.pappinen@uef.fi; 2Natural Resources Institute Finland (Luke), 80100 Joensuu, Finland; aki.sinkkonen@luke.fi (A.S.); pertti.pulkkinen@luke.fi (P.P.); 3Biological and Environmental Sciences, University of Helsinki, 00014 Helsinki, Finland; ruhui.wen@helsinki.fi

**Keywords:** hybrid aspen, European aspen, survival, growth, creosote oil, diesel oil, density

## Abstract

This study was conducted to assess the survival rates, growth, and chlorophyll fluorescence (Fv/Fm) of four hybrid aspen (14, 191, 27, 291) and two European aspen (R3 and R4) clones cultivated in creosote- and diesel oil-contaminated soil treatments under three different plant densities: one plant per pot (low density), two plants per pot (medium density), and six plants per pot (high density) over a period of two years and three months. Evaluating the survival, growth, and Fv/Fm values of different plants is a prerequisite for phytoremediation to remediate polluted soils for ecological restoration and soil health. The results revealed that contaminated soils affected all plants’ survival rates and growth. However, plants grown in the creosote-contaminated soil displayed a 99% survival rate, whereas plants cultivated in the diesel-contaminated soil showed a 22–59% survival rate. Low plant density resulted in a higher survival rate and growth than in the other two density treatments. In contrast, the medium- and high-density treatments did not affect the plant survival rate and growth to a greater extent, particularly in contaminated soil treatments. The effects of clonal variation on the survival rate, growth, and Fv/Fm values were evident in all treatments. The results suggested that hybrid aspen clones 14 and 291, and European aspen clone R3 were suitable candidates for the phytoremediation experiment, as they demonstrated reasonable survival rates, growth, and Fv/Fm values across all treatments. A superior survival rate for clone 291, height and diameter growth, and stem dry biomass production for clone 14 were observed in all soil treatments. Overall, a reasonable survival rate (~75%) and Fv/Fm value (>0.75) for all plants in all treatments, indicating European aspen and hybrid aspen have considerable potential for phytoremediation experiments. As the experiment was set up for a limited period, this study deserves further research to verify the growth potential of different hybrid aspen and European aspen clones in different soil and density treatment for the effective phytoremediation process to remediate the contaminated soil.

## 1. Introduction

Polycyclic aromatic hydrocarbons (PAH) have been identified as hazardous pollutants globally due to their toxic, mutagenic, and carcinogenic properties and prolonged biodegradation rate [1]. Harmful levels may persist in the soil for a long time. It can be released into water, air, and soil by wood and coal combustion, diesel and petrol combustion, mining, and industrial processes, posing a severe threat to the ecosystem and human health [2,3]. It has been estimated that organic contaminants represent 55% of all contaminants in the EU [4]. Approximately 24% of European soils have been affected by mineral oil contamination [5]. In Finland, organic contaminants are also considered one of the significant sources of soil pollution [6], whereas 8% of the sites are contaminated with PAH [7]. The PAH and total petroleum hydrocarbons (TPH) contaminants pollute surface water, groundwater, soil, and sediments, which are threats to the ecosystem and urgently need to be solved [8]. To overcome the problems associated with environmental pollution, several researchers have proposed an environmentally friendly and cost-effective solution called phytoremediation—the manipulation of plants to reduce pollutant toxicity levels and remove or immobilize toxic pollutants from the soil through plants [9,10]. The limitations of phytoremediation include various factors such as plant species and the toxicity level of contamination [11]. The number of pollutants that accumulate in leaves, shoots, and roots or are removed by plant organs has been shown to vary considerably between clones and soil treatments [12,13]. Due to the different physiological and chemical properties of trees, some plant species accumulate a significant amount of particular PAH while excluding others [13,14]. It is crucial to identify the appropriate plant species to remediate hydrocarbons–contaminated sites effectively. According to Sivaram et al. [14], selecting suitable plant candidates is the primary determinant of the success of phytoremediation studies. We hypothesized that variation in the growth of European aspen and hybrid aspen clones may occur in different soil treatments. Evaluating the survival, chlorophyll fluorescence parameters, and growth development of aspen trees in contaminated soils are prerequisites for establishing phytoremediation experiments [13,15]. Li et al. [16] and Salam et al. [17] reported that plant growth was positively correlated with successful phytoremediation.

Biomass production relies heavily on photosynthesis, which allows for the study of phytoremediation [18]. Chlorophyll fluorescence (Fv/Fm, the maximum quantum yield of PSII photochemistry) is a crucial parameter to assess plant tolerance to pollutants and identify potential plants for phytoremediation [19,20]. A decrease in chlorophyll fluorescence is caused by a reduction in photosynthetic efficiency owing to stress conditions such as heat, drought, herbicide damage, disease, pollutants, and nutrient deficiency [21,22]. Phytoremediation experiments using European and hybrid aspen clones in Finland have not produced any studies on photosynthetic activity. This study may help obtain the basic knowledge of photosynthetic activity to select the best aspen clone for phytoremediation experiments in a boreal climate.

The European aspen (*Populus tremula* L.) is one of the most widely distributed trees in the world [23]. It is native to Finland and often grows in mixed stands of pine, spruce, and birch [24]. The hybrid aspen is a cross between the European aspen and the North American trembling aspen (*P. tremuloides* Michx.) [25]. European aspen and hybrid aspen belong to the *Salicaceae* family. They are specially adapted to boreal climates and exhibit significant genetic heterogeneity [26]. European aspen, especially hybrid aspen, have some desirable properties for which they have the potential to increase their importance in forestry in the future [25,27]. European aspen and hybrid aspen are essential to tree species for forest biodiversity. Therefore, they are recommended to be left growing in commercial forests [28].

European aspen and hybrid aspen have some unique characteristics that are suitable for phytoremediation experiments, such as quick physiological responses to environmental factors, ease of establishment, fast-growing with a height of 15–50 m, and deep root systems [23,26,29,30]. Fast-growing plant species with high biomass and vigorous root systems are advantageous for microbial growth and contaminant degradation due to their increased root surface area [17,31]. A typical feature of native and hybrid aspens is the vigorous production of root suckers. This feature can be effectively applied in forest management for biomass production [32,33]. Studies on landfills have demonstrated the ability of European aspen and hybrid aspen to grow, take up, and tolerate pollutants [34,35]. In another phytoremediation study, *Populus* plants were used for leachate and wastewater treatments, and they generally showed better growth and resistance than the *Salix* plants [36]. Recently, Salam et al. [37] noted that European aspen and hybrid aspen have the potential to survive and grow in hydrocarbon-polluted soils. Marmiroli et al. [38] and Salam et al. [39] reported that the removal percentage of pollutants depends on the plant species and plant growth. In a phytoremediation study, Van Dillewijn et al. [40] reported that transgenic hybrid aspens have considerable potential for remediating contaminated soil and underground aquifers.

The planting density is an important parameter for evaluating plant growth during phytoremediation. Plant density can affect the growth and quality of plants in several ways [41]. Nutrient availability may depend on the plant species and density [42]. Plant growth and quality are the prerequisites for phytoremediation. In a phytoremediation study, plant density was found to affect plant growth parameters, such as survival rate, height, diameter, and biomass [7]. In a slight contradiction to the advancements, systematic studies incorporating the importance of plant density to the phytoremediation scheme are still quite limited, despite considerations and studies showing the need to grow plants at an optimal or maximal plant density [43,44]. The optimal density for soil or water remediation is seldom the same as that in conventional agriculture, where yield per hectare is the priority [45]. Biomass per plant is the highest at intermediate plant densities, and plant biomass per unit area increases with increasing plant density until a specific density is reached [46,47,48]. However, the effects of toxins on plant survival at high plant densities are unknown. We also posit that different plant densities may affect the survival and growth of aspen clones in the soil treatments.

We aimed to provide an environmentally friendly solution for the effective remediation of PAH–contaminated areas using suitable European aspen and hybrid aspen clones and to identify future research needs for their efficient use. However, despite having suitable characteristics for phytoremediation and a wide range of European and hybrid aspen applications, there is minimal information in the literature on the growth assessment of these species and their potential to remediate polluted soils. Moreover, to the best of our knowledge, the growth of European aspen and hybrid aspen clones based on plant density when exposed to PAH–contaminated soils has not been previously reported. The main aim of this study was to assess the effects of contaminated soil on the growth attributes of European aspen and hybrid aspen clones at different plant densities. Improving the knowledge of hybrid aspen behavior in stressful environments will optimize the implementation of accurate-scale phytoremediation field trials.

## 2. Materials and methods

### 2.1. Soil Materials

The soil used in this study was collected from a former wood treatment area (~7 ha) located in Somerharju, southeastern Finland (60°92′ N, 27°56′ E, Luumäki) (Figure 1). During the growing season, the mean monthly temperature in the study area ranges from 21 to 25 °C in the summer (May–August) and from −10 to 22 °C in the winter (November-March). The length of daylight varies from 5 h in winter to 20 h in summer [37]. The creosote preservation facility for railway sleepers was situated in the central part of the study area (~1.2 ha) from 1947 to 1958 [37]. During this time, an extensive leak occurred in this area, causing 10,000 L of creosote to spill into the soil [49]. PAHs and TPHs were detected at elevated levels in 3.0 ha of soil and 4.0 ha of groundwater. The elevated PAH and TPH deposits near the groundwater are approximately 10–11 m below the soil surface. Up to 580,000 µg L^−l^ PAH was found in the groundwater [7,49]. Thus, the central part of the site is highly polluted with creosote [49]. About 20–40% of creosote comprises 16 (United States Environmental Protection Agency (USEPA) defined) priority PAH pollutants [50].

In 2013, creosote soil was taken at a depth of 0 to 50 cm from the Somerharju trial square site H14 (Appendix A). The square area was 400 m^2^, and the creosote soil used in our experiment was collected from the four corners and middle of the square site H14. Creosote soil was used in this experiment to serve as soil contaminated with hydrocarbons. The physical and chemical properties of creosote soil collected from the study site are presented in Table 1. In general, the soil texture was classified as sandy, with mostly medium-and fine-sand particles. To determine the soil properties, soil samples were collected from severely and less contaminated areas. Pristine soil was randomly collected at a depth of 0 to 50 cm from the uncontaminated area to serve as control soil. The uncontaminated area was located approximately 400–500 m away from the contaminated area. Finally, the pristine soil was spiked with fresh diesel oil (collected from a local Läyliäinen Neste station) at the rate of 5 mg g^−1^ DW soil by injecting a 50 mL syringe in the concrete mixer to serve as diesel-contaminated soil. Approximately 30–40 kg of dry soil were used in the concrete mixer per round, and the duration of mixing was 2–3 min per round to combine dry soil with fresh diesel oil.

### 2.2. Plant Materials

We used two European aspen and four hybrid aspen clone seedlings. In August 2012, these seedlings were produced using the micropropagation method [51] in the Haapastensyrjä unit laboratory. Parent plants were collected from several aspen trials in southern Finland. After multiplication, the clones began rooting in January 2013. The seedlings were transplanted into boxes at room temperature in February 2013 and kept in the boxes until June 2013. Finally, seedlings were planted in pots for the greenhouse experiment, and their height ranged from approximately 0.5 cm to 15 cm depending on the clone.

### 2.3. Experimental Design

The experiment was set up in a plastic greenhouse at the Haapastensyrjä tree-breeding center in Southern Finland (60°36′ N, 24°25′ E) for a period of two years and three months from June 2013 to October 2015. A total of 486 trees were transplanted at three planting densities: one plant per pot (coded as low density), two plants per pot (coded as medium density), and six plants per pot (coded as high density) in 162 10 L plastic pots (height: 24 cm; top surface diameter: 26 cm; bottom surface diameter: 21 cm), which were filled with ~7–8 kg of soil consisting either of control soil or contaminated soil. Sampled soils have not been sorted, ground, or sieved. The treatments for the soil in the experiment were as follows: new clean soil (coded as control), creosote soil polluted with PAH and TPH (coded as old creosote), and pristine soil spiked with fresh diesel oil (coded as new diesel). The three planting densities were selected to represent the level of competitive stress in a typical growth environment. Altogether 486 trees were obtained from two European aspen clones (coded as R3, R4) and four hybrid aspen clones (coded as 14, 27, 191, 291). Each pot had a small hole at its bottom to ensure proper drainage. Plants were placed on separate plates to avoid cross-contamination. Each pot was placed in a basin to avoid the loss of contaminants when the plants were subjected to irrigation. The greenhouse experiment was designed with three blocks corresponding to replicates to test the following three main factors: the effect of soil contamination (creosote, diesel, and uncontaminated control), planting density (low, medium, and high), and aspen clonal variation. Thus, we used a randomized factorial block design with three treatments for the soil, three plant densities, and six clones of European aspen and hybrid aspen, resulting in 54 pots, 162 individual plants per block, and a total of 486 trees in the three blocks (Appendix A). We did not repeat any treatment during the single treatment block. The placement of each pot inside the replicate was randomized to overcome the microclimate effect induced by its location in the greenhouse. In 2013 and 2014, plants were placed in non-heated storage around October and moved back to a greenhouse again in May, when buds presented new activity (Appendix A). The temperature in the storage was maintained at 22–24 °C with a photoperiod of 16 h light at 20 °C and 8 h dark at 19 °C, and relative humidity at around 80%. An irrigation volume of 0.5 L per pot was added to each pot twice a week during the summer period (June, July, and August) and once a week during the rest period, except for the dormancy period.

Survival was analysed in all blocks at the end of the third growing season (September 2015). Plants were determined to be dead only when all of the following phenomena occurred during the entire growing season. First, none of the stems had any green leaves. Second, the plants did not recover from the buds or suckers. Third, the plant height did not change. Absolute height was estimated by subtracting the initial height in September 2013 from the final height in September 2015. The diameter was measured at the end of the third growing season for each tree at the height of approximately 5 cm. The measurement was carried out at 0.01 mm accuracy with a sliding rule measurement tool (Vernier Caliper of Storm instruments). The biomass of the plants in one randomly selected block was measured at the end of the third growing season (September 2015). The stems were harvested and washed carefully to remove soil and dust from the dry stem biomass. The samples were then dried at 105 °C for 24 h and cooled at room temperature for 12 h in a drying vessel prior to biomass determination. The measurement was determined by each pot and was accomplished by a weighted instrument with an accuracy of 0.01 g.

At the end of the experiment, three healthy and fully developed leaves from each treatment were collected to estimate chlorophyll fluorescence (Fv/Fm). A portable fluorometer (PAM 2100 Walz, Effeltrich, Germany) was used to estimate the chlorophyll fluorescence using the protocols described by Maxwell and Johnson [21] and Evlard et al. [52]. The chlorophyll fluorescence (Fv/Fm) of the plants was calculated as (Fm − Fo)/Fm, where Fv indicates the maximum variable fluorescence emission, Fm represents the maximum fluorescence, and Fo represents the minimum fluorescence [21].

### 2.4. Statistical Analysis

“R Statistical Software (v4.1.2; R Core Team [53])” was used for statistical analysis. We used a generalized linear mixed model or generalized linear model to analyze the deviance table and determine whether factors such as treatment, clone, and density significantly affected survival rates and growth parameters (*p* < 0.05). A generalized linear mixed model or generalized linear model was used to compare treatments in the least-squares means. When comparing pairs of treatments, Tukey’s test was adjusted for *p*-values. The generalized linear mixed model or generalized linear model used for ANOVA was as follows:

Model 1: survive~treatment + clone + density + (1|block)

Model 2: absolute height~diameter + treatment + clone + density + (1|block)

Model 3: diameter~absolute height + treatment + clone + density + (1|block)

Model 4: stem dry biomass~treatment + clone + density

The Kruskal-Wallis test was used to determine the significant effects of clone, density, and soil treatment on the Fv/Fm value (*p* < 0.05). The Wilcoxon signed-rank test was performed to compare the significant differences between the Fv/Fm values of the two clones.

## 3. Results

### 3.1. Survival Scenario

At the end of the experiment, 368 trees were alive, with a survival rate of 76% across all treatments. A 100% survival rate was recorded for hybrid aspen clone 291 and European aspen clones R3 and R4 under control soil. In general, plants grown in diesel oil-contaminated soil treatment displayed a 22–59% survival rate, whereas plants grown in old creosote-contaminated soil treatment showed a 99% survival rate (Figure 2a). The plants cultivated in low-density (one plant per pot) and medium-density (two plants per pot) treatments had a 10% higher survival rate than those cultivated in high-density treatment (six plants per pot) (Figure 2b). Soil treatment, density, and the clone had a significant effect (*p* < 0.05) on the survival rate (Table 2). Significant differences (*p* < 0.05) in survival rates between all soil treatments were also evident.

### 3.2. Absolute Height

Contaminated soil treatments reduced the growth compared to control in all clones (Figure 3a). The diesel-contaminated soil treatment decreased absolute height by 5–33% for hybrid aspen clones (14, 191, 27, and 291) and 28–44% for European aspen clones (R3 and R4) when compared to their respective controls. Creosote-contaminated soil treatment also decreased absolute height by 19–38% for hybrid aspen clones and 20–22% for European aspen clones compared to their respective controls (Figure 3a). Superior growth was apparent in clone 14 compared to other hybrid aspen clones grown in all soil treatments, whereas clone R3 was superior to R4 in growth when plants were cultivated in creosote-contaminated and controlled soil treatments. In contrast, clone R4 was superior to R3 in terms of growth when plants were grown in diesel-contaminated soil treatment (Figure 3a). A decreasing trend in the growth of the plants was evident from the low-density treatment to the high-density treatment (Figure 3b). Significantly higher (19–33%) growth was observed in the plants cultivated in low-density treatment than plants cultivated in medium- and high-density treatments. Diameter, soil treatment, density, and the clone had significant effects (*p* < 0.001) on absolute height (Table 2).

### 3.3. Diameter

Among all hybrid aspen clones, clone 14 had the largest diameter in the control soil treatment (5.69 cm). Contrary, clone 291 had the smallest diameter in the creosote-contaminated soil treatment (2.98 cm). In European aspen clones, clone R3 showed the largest diameter in the control soil treatment (5.42 cm), whereas R4 showed the lowest diameter in the diesel-contaminated soil treatment (3.14 cm) (Figure 4a). Overall, plants treated in control showed more growth than plants treated in the contaminated soil across all clones (Figure 4a). Diesel-contaminated soil reduced the diameter by 6–11% in hybrid aspen clones (14, 191, 27, 291) and 20–39% in European aspen clones (R3 and R4) compared to their respective controls. The creosote-contaminated soil treatment also reduced in diameter by 20–30% in hybrid aspen clones and 12–23% in European aspen clones compared to their respective controls. As compared with clone 191 grown in control soil, its diameter declined by 20% in contaminated soil (Figure 4a). In low-density treatment, plant diameter increased by 11% compared to plants of medium-density treatment and by 20% compared to plants of high-density treatment (Figure 4b). Height, soil treatment, and the clone had a significant impact (*p* < 0.001) on diameter growth, whereas density did not have a significant influence (*p* = 0.4893) (Table 2).

### 3.4. Dry Biomass of Stem

In control soil, clone R3 produced the greatest amount of dry biomass, followed by clone 27 (4.21 and 3.76 g pot^−1^), respectively. In diesel-contaminated soil, clone 14 produced the highest dry biomass, followed by clone 291 (2.53 and 0.82 g pot^−1^), respectively. In creosote-contaminated soil, clones 14 and 291 produced the greatest dry biomass, followed by clone R3 (1.85, 1.85, and 1.69 g pot^−1^), respectively (Figure 5a). Plants of diesel-contaminated soil treatment produced 9–93% less biomass than plants of control soil treatment across all clones. Plants of creosote-contaminated soil treatment also produced 34–63% less biomass than plants of control soil treatment. Plants cultivated in creosote-contaminated soil treatment produced 31–465% more biomass than plants cultivated in diesel-contaminated soil treatment for all clones, whereas clone 14 grown in creosote-contaminated soil treatment produced 27% less biomass than clone 14 grown in diesel-contaminated soil treatment (Figure 5a). Plants in low-density treatment exhibited the highest dry biomass, followed by medium-density and high-density treatments (4.59, 2.12, and 1.32 g pot^−1^), respectively. Plants grown in low-density treatment produced significantly more 53–71% biomass than plants grown in medium-and high-density treatments (Figure 5b). Soil treatment and density significantly affected dry stem biomass production (*p* < 0.001). In contrast, the clones did not significantly affect stem dry biomass production (*p* = 0.458) (Table 2).

### 3.5. Chlorophyll Fluorescence (Fv/Fm)

Hybrid aspen clones grown in contaminated soil showed slightly higher Fv/Fm values than those grown in the control. The European aspen clone R4 had a higher Fv/Fm value in the diesel-contaminated soil treatment than in the control and creosote-contaminated soil treatments. In contrast, European aspen clone R3 showed a higher Fv/Fm value in the control soil treatment than in the contaminated soil treatments. However, no significant differences were found among clone Fv/Fm values in all treatments, except for the R4 clone (Figure 6a). Plants of low-density treatment slightly reduced the Fv/Fm values compared to plants of medium-and high-density treatment (Figure 6b). The results of the Kruskal-Wallis test suggested that clones had a significant effect on Fv/Fm value (χ^2^ = 26.446, df (degree of freedom) = 5, *p* < 0.001), whereas density (χ^2^ = 1.1435, df = 2, *p* = 0.5645) and soil treatments (χ^2^ = 2.3456, df = 2, *p* = 0.3095) did not have a significant effect on Fv/Fm value.

## 4. Discussion

European and hybrid aspen clones survived well with resistance to stress in all treatments. This could be due to the fact that when plants grow in polluted soils, phenolic compounds and non-enzymatic and enzymatic antioxidants facilitate the activation of a tolerance mechanism and produce a robust antioxidant defense system against pollutant stress [17]. In soils polluted with hydrocarbons, plants often experience metabolic changes, such as hormone production and enzyme synthesis, which lead to substrate modification and higher enzyme levels [54]. Zalensy et al. [8] noted that ~a 75% survival rate for plants grown in soils contaminated with hydrocarbons would indicate that plants are suitable for successful phytoremediation. This would suggest that European aspen and hybrid aspen clones have considerable potential for phytoremediation experiments since European aspen and hybrid aspen clones showed a 76% survival rate in this study. A field trial demonstrated that European aspen and hybrid aspen clones could survive in hydrocarbons-contaminated soil with a 59–64% survival rate [37]. Compared to the other clones, hybrid aspen clone 291 had a higher survival rate in diesel-contaminated soil (Figure 2a). These results are consistent with the findings of Salam et al. [37], who reported the survival of European aspen and hybrid aspen clones grown in hydrocarbons-contaminated soils. In diesel-contaminated soil, European aspen clone R4 survived more effectively than R3 (Figure 2a). Contrary to this result, Salam et al. [37] documented that European aspen clone R3 survived more effectively than R4 grown in PAH- and TPH-contaminated soil. Plants exposed to creosote-contaminated soil exhibited higher survival rates than those exposed to diesel-oil-contaminated soil. Clonal variations in the survival rates were evident in the control and diesel oil-contaminated soil treatments (Figure 2a). This discrimination can be ascribed to different clone types, growth media, and exposure times. The differences observed in survival rates between clones could also be due to the different acclimation capacities shared between subpopulations [55]. These results align with the findings of Camille [7], who reported the survival of different aspen and hybrid aspen clones grown in fresh diesel oil- and creosote-contaminated soils under different stand densities. Light oils can cause acute injuries (oils with benzene, toluene, and xylene) and heavy oils can cause chronic injuries (oils with PAH). Fresh crude oil is more toxic than old oil. Oil can provoke physical interference with the gaseous exchange if stomata are blocked, a loss of energy in heat due to a useless increase in oxygen uptake, penetration, damage in mitochondria, and inhibition of essential transport mechanisms [56]. The reduced survival of the plant in the soil with the new diesel oil can be explained by these effects, as lower survival rates were observed (Figure 2a). Aging, caused by the sequestration of compounds bound to soil particles, decreases pollutant toxicity and facilitates higher survival rates of plants grown in old creosote-contaminated soils [57]. In the contaminated soil treatments, the plants exposed to high-density treatment showed a lower survival rate than plants exposed to other density treatments (Figure 2b). Plant density did not change survival in the contaminated soil treatments, whereas survival in pristine soil became patchy as plant density increased from two to six plants per pot. According to Qin et al. [58], high planting densities could increase the mortality of a species owing to intraspecific competition, resulting in a low survival rate.

Biometric analysis (e.g., height, diameter, and biomass) can be considered an essential tool for evaluating pollutant tolerance and the production capacity of plants grown in contaminated soils [17]. The contaminated soil treatments resulted in lower growth parameter values, such as absolute height, diameter, and stem dry biomass, in the plants than in the control treatments (Figure 3a, Figure 4a, Figure 5a). Garnica et al. [59] also noted that plants grown in hydrocarbon-contaminated soil treatments showed lower growth and produced less dry stem biomass than plants grown in the control. A similar result was reported by Camille [7]. This may be due to nutrient deficiency, pollutant toxicity, pollutant stress, low organic matter content, stomatal limitations, photoinhibition, reduction in chlorophyll content, or enzyme activities involved in CO_2_ fixation that retard plant metabolic, physiological, and biochemical processes [60]. In phytoremediation studies, Nguemté et al. [61] and Shirdam et al. [62] reported that toxic compounds can inhibit plant growth in the presence of total petroleum hydrocarbons.

The highest growth was observed for hybrid aspen clone 14 compared to other hybrid aspen clones subjected to contaminated soils, while the highest growth was observed for European aspen clone R3 compared to clone R4 subjected to contaminated soils (Figure 3a, Figure 4a and Figure 5a). In a similar study, the greatest height and diameter were reached in hybrid aspen clone 14 compared to other clones in all densities and contaminated soil treatments [7]. Garnica et al. [59] reported that hybrid aspen clones had better adaptability than European aspens grown in creosote-contaminated soil. In a field trial, hybrid aspen clone 14 had the highest growth in hydrocarbon-contaminated soil compared to the other clones, while European aspen R3 had the highest growth compared to European aspen clones R4 and R2 [37]. The height, diameter, and biomass of European aspen clones grown on creosote-contaminated soil were greater than those grown in diesel oil-contaminated soil (Figure 3a, Figure 4a and Figure 5a). Owing to aging or weathering, soil with old diesel contamination is less toxic, which helps it survive better and further facilitates better growth than new diesel contamination [57]. Diesel oil-contaminated soil had the greatest height for hybrid aspen clones 14 and 291, the greatest diameter for hybrid aspen clones 14, 27, and 291, and the greatest stem dry biomass production for clone 14 compared to those grown in creosote-contaminated soil (Figure 3a, Figure 4a and Figure 5a). This could be due to the fact that when plants were grown in diesel oil-contaminated soil in 2013, the soil became less toxic owing to weather processing in 2015, which aided better plant growth, indicating that the general effect of soil contamination on growth decreases with time [7]. Furthermore, clones may be better adapted to new environments and to diesel oil pollutant stress. Differences in growth and dry biomass production among clones were observed across all treatments. These results were consistent with the findings of Camille [7]. Plant physiology and biochemistry may have led to the varying acclimation capacities. This may have caused the observed differences in growth between clones [17]. The variation in plant growth may also be due to the different growth media, nature of pollutants, level of pollutant concentrations, exposure, and cultivation time.

To achieve consistently high remediation efficiency, planting density should be considered [63]. In general, plants in the low-density treatment showed higher plant growth parameter values than those in the other two density treatments (Figure 3a, Figure 4a and Figure 5a), although high- and medium-density effects on height and diameter were evident to a lesser extent, particularly in the contaminated soil. Several studies have documented that low plant density results in fast plant growth and vice versa [58,64,65]. Planting high densities could increase the death rates of species owing to intraspecific competition for resources, leading to lower plant growth [58]. Jiang et al. [65] and Liu et al. [66] also reported that the decline in plant growth in higher density treatments might be due to competition for resources and space among plants grown in hydrocarbon-contaminated soil. In the phytoremediation experiment, the average root length and diameter increased with increasing plant density, whereas the average root surface area per plant decreased. Low-density treatments aid faster plant growth due to the high efficiency of nutrient acquisition [64,67]. Liu et al. [66] concluded that a lower plant density would be a better choice for phytoremediation of PAHs based on the sustainability of the ecosystem. Interactions between plants and other organisms such as microbes, pathogens, and herbivores may change resource availability and allocation, and site properties are central to plant growth. Contrary to our results, several theoretical and empirical laboratory and field studies have concluded that toxins decrease or prevent plant growth at low densities. Exceptions to this rule may be due to toxin release from plant residues [68] and plant autotoxicity [69]. In addition, the level of size asymmetry and stimulatory effects of low toxin doses may alter the way neighboring plants share resources and take up toxins [70,71,72,73,74,75], which may change mortality in a plant stand. This discrimination may be due to density treatments, types of pollutants, growth factors, variation in plant species, and rotation period.

Chlorophyll fluorescence (Fv/Fm) is used to evaluate stress on plant metabolism [52], and a value <0.75 indicates that a tree has low photosynthetic efficiency [76]. In this study, all of the clones exhibited good photosynthesis efficiency since the Fv/Fm values ranged from 0.75 to 0.78 across all treatments (Figure 6a,b), and the results agree with findings reported by other studies (e.g., [7,17,77]). This suggests that all clones used in this study were not stressed and were suitable for phytoremediation. Productive photosynthesis enhances the capability of plants to produce metabolites for fortification and growth [78]. The trees with the best photosynthetic efficiency, and thus the best Fv/Fm values, were expected to present better growth, as biomass production is linked to photosynthetic efficiency. Furthermore, trees with better internal force to resist outside constraints (best PI) should be better able to resist stress conditions, and thus allocate their resources for growth. For instance, the growth was satisfactory in clones 14 and 27, indicating good chlorophyll fluorescence values. A lower Fv/Fm value was found in plants in the low-density treatment in comparison to plants in the medium- and high-density treatments (Figure 6b). This could be explained by the low-stress conditions experienced by plants in the low-density treatment, or perhaps plants didn’t need to fully utilize photosynthetic activities for growth [7]. Oil is thought to inhibit gaseous exchange and damage the mitochondria [56]. Previous experiments have shown that the photosynthetic rate is influenced by diesel contamination and that stomatal conductance is inhibited in most species [7].

The main limitation of this experiment was that the plants were cultivated in different soil treatments for a limited time (two years, three months). It is difficult to select the best clone for phytoremediation experiments within this limited time frame. It has been estimated that three to four years is sufficient for clonal selection of phenotypes in aspen, according to Stener and Karlsson [79]. In addition, to consider possible canker apparition and determine specific industrial value, plants should not be selected before 10–15 years of age in the field [79]. For reliable results, field trials of 10 years are expected. The hybrid aspen and European aspen clones were too few to qualify as being superior. This study was also limited by the absence of soil amendment application to stimulate plant survival and growth. This is a preliminary experiment to test the phytoremediation parameters, and the removal percentages of hydrocarbons by different clones were missing owing to a lack of budget. Considering these factors, further research is needed to explore the growth potential of different aspen clones at different densities and soil treatments in phytoremediation experiments.

## 5. Conclusions

Hybrid aspen clones exhibited higher survival and growth rates than European aspen clones in all treatments, except for the Fv/Fm value. In hybrid aspen, clone 291 had the highest survival rate, followed by clone 14, whereas clones 191 and 27 were not as good in general for the survival rate. All clones studied in creosote-contaminated soil had excellent survival rates. European aspen clones R3 and R4 showed considerable potential for a survival rate in control soil. Overall, hybrid aspen clone 14 could be advocated for phytoremediation since it displayed an acceptable survival rate, growth, and Fv/Fm value, followed by 291. A reasonable survival rate, growth, biomass production, and healthy Fv/Fm value were also evident for European aspen clone R3, recommending that clone R3 could also be a phytoremediation experiment option. Poor growth and lower Fv/Fm values were recorded in the plants grown in contaminated soil treatments than those grown in control soil treatment. In all soil treatments, plants subjected to low-density treatment demonstrated a good survival rate and growth. Results indicate that clones 14, 291, and R3 are ideal candidates to remediate hydrocarbons-polluted soil since they had reasonable survival rates, growth rates, and higher Fv/Fm values.

## Figures and Tables

**Figure 1 plants-11-01970-f001:**
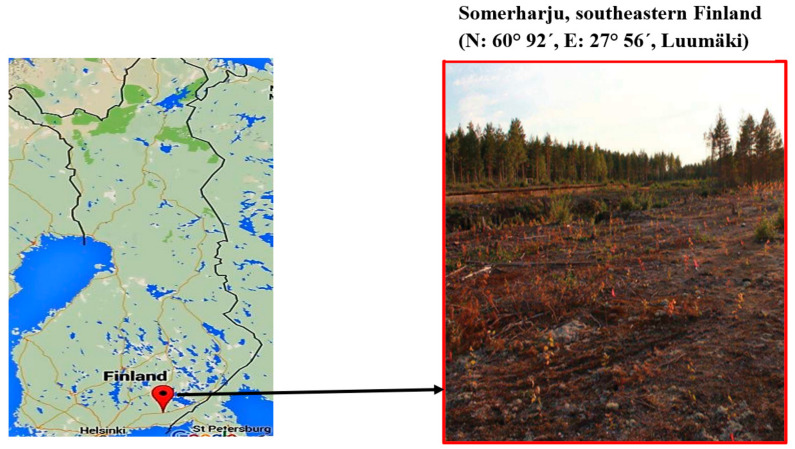
Map of the study area Somerharju.

**Figure 2 plants-11-01970-f002:**
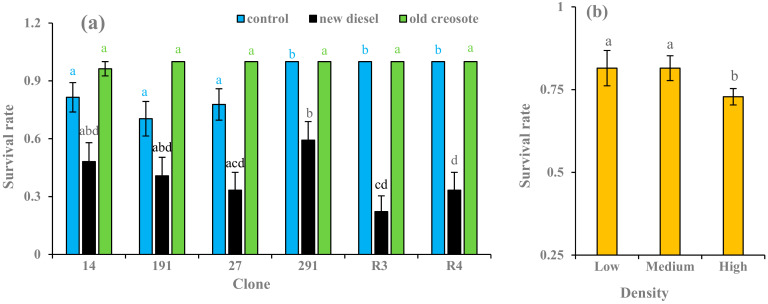
Average (*n* =3) survival rates of European aspen and hybrid aspen clones grown in different soil treatments (**a**) and plant densities (**b**). Error bar indicates ± standard error (SE). Means between clones followed by the same lower-case letters with the same font color are not significantly different (*p* > 0.05). Means between density treatments followed by the same lower-case letters are also not significantly different (*p* > 0.05).

**Figure 3 plants-11-01970-f003:**
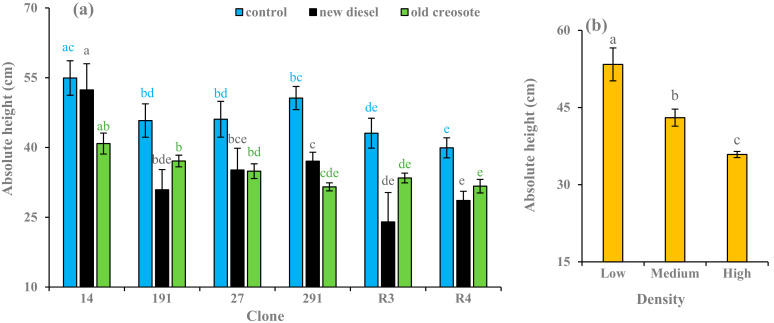
Average (*n* = 3) absolute height (cm) of European aspen and hybrid aspen clones grown in different soil treatments (**a**) and plant densities (**b**). Error bar indicates ± standard error (SE). Means between clones followed by the same lower-case letters with the same font color are not significantly different (*p* > 0.05). Means between density treatments followed by the same lower-case letters are also not significantly different (*p* > 0.05).

**Figure 4 plants-11-01970-f004:**
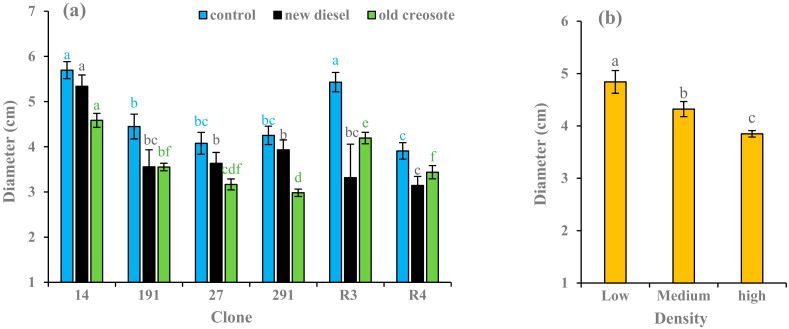
Average (*n* = 3) diameter (cm) of European aspen and hybrid aspen clones grown in different soil treatments (**a**) and plant densities (**b**). Error bar indicates ± standard error (SE). Means between clones followed by the same lower-case letters with the same font color are not significantly different (*p* > 0.05). Means between density treatments followed by the same lower-case letters are also not significantly different (*p* > 0.05).

**Figure 5 plants-11-01970-f005:**
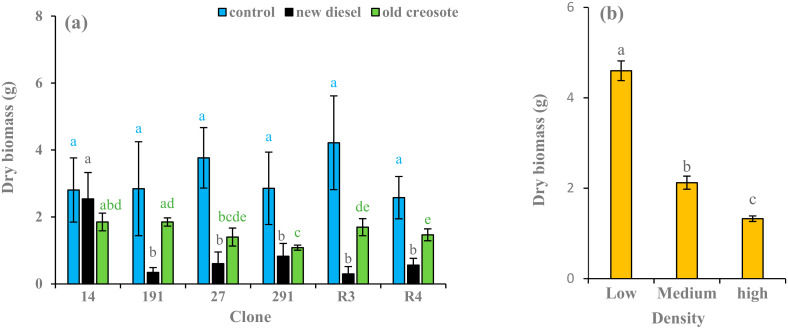
Average (*n* = 3) dry biomass of stem (g) of European aspen and hybrid aspen clones grown in different soil treatments (**a**) and plant densities (**b**). Error bar indicates ± standard error (SE). Means between clones followed by the same lower-case letters with the same font color are not significantly different (*p* > 0.05). Means between density treatments followed by the same lower-case letters are also not significantly different (*p* > 0.05).

**Figure 6 plants-11-01970-f006:**
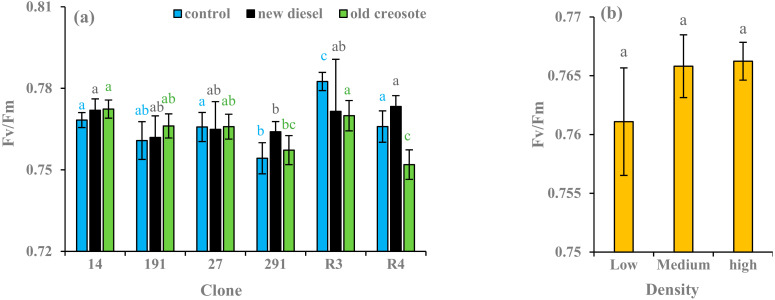
Mean Fv/Fm values (*n* = 3) of European aspen and hybrid aspen clones grown in different soil treatments (**a**) and plant densities (**b**). Error bar indicates ± standard error (SE). Means between clones followed by the same lower-case letters with the same font color are not significantly different (*p* > 0.05). Means between density treatments followed by the same lower-case letters are also not significantly different (*p* > 0.05).

**Table 1 plants-11-01970-t001:** Description of soil pollutants (total PAHs, 16 USEPA priority PAHs, TPH (C10–C21, C22–C40, and C10–C40)), nutrients, and soil texture.

Properties of Soil	Unit	Soil Layer						References
		0–10 cm				5–10 cm	10–50 cm	
		Minimum	Maximum	Mean	SE			
		*n* = 4						
pH		5 **a**	7.8 **a**	5.94	0.31	6.1 **b**	6.1 **b**	[37,49,50]
		*n* = 55						
Total PAHs	ppm	0.16	714	96.51	24.1	532 **b**	164 **b**	
Naphthalene		0.01	2.19	0.15	0.04			
Acenaphthylene		0.01	3.10	0.37	0.08			
Acenaphthene		0.01	7	0.23	0.13			
Fluorene		0.01	11	0.46	0.23			
Phenanthrene		0.01	22	0.87	0.42			
Anthracene		0.01	160	7.13	3.87			
Fluoranthene		0.01	213	21.89	6.09			
Pyrene		0.01	224	24.25	6.50			
Benz(a)anthracene		0.01	376	13.4	6.93			
Chrysene		0.01	100	11.35	2.86			
Benzo(b)fluoranthene		0.01	47.4	7.15	1.5			
Benzo(k)fluoranthene		0.01	22.9	4.12	0.87			
Benzo[a]pyrene		0.01	21	3.04	0.65			
indeno(1,2,3-cd)pyrene		0.01	5.62	0.98	0.18			
Benzo[ghi]perylene		0.01	5.61	0.67	0.14			
Dibenz[a.h]anthracene		0.01	2.26	0.44	0.08			
C10–C21		10	580	99.84	20.44	556 **b**	190 **b**	
C22–C40		10	1780	338.8	67.06	2120 **b**	688 **b**	
C10–C40		20	2350	437.1	85.79	2675 **b**	875.5 **b**	
		*n* = 30						
C	%	0.05	1.46	0.41	0.08			
N		0.01	0.06	0.03	0.00			
Ca	(mg kg^–1^)	849	3020	1272	81.84			
Cu		2.91	20.7	6.71	0.98			
Fe		4490	6950	5204	128.7			
K		393	690	510.6	13.37			
Mg		513	924	613.4	21.45			
Ni		1.25	5.69	2.10	0.21			
P		149	326	207.6	7.35			
Pb		2.81	15.4	5.07	0.56			
Zn		13.8	31.2	19.30	0.92			
coarse sand	%	2.03	28.74	10.53	1.29			
medium sand		28.85	59.41	42.53	1.37			
very fine sand		18.26	52.55	40.06	1.66			
Slit		1.01	13.86	6.15	0.69			
clay		0.31	1.16	0.73	0.05			

**a**: pH in all blocks (I–IV) of the study area [49]. **b**: hydrocarbon concentrations in square site H14 (block II) of the study area [37].

**Table 2 plants-11-01970-t002:** Analysis of deviance (ANOVA).

Variable	Factors	χ^2^	DF (Degree of Freedom)	*p*-Value	Model
survival rate	clone	l2.3	5	0.0312	1
	soil treatment	93.2	2	<0.001	
	density	7.2	2	0.0272	
absolute height	clone	106.99	5	<0.001	2
	soil treatment	15.05	2	<0.001	
	density	35.65	2	<0.001	
	diameter	423.22	1	<0.001	
diameter	clone	250.68	5	<0.001	3
	soil treatment	20.88	2	<0.001	
	density	1.43	2	0.4893	
	height	425.7	1	<0.001	
stem dry biomass	clone	4.660	5	0.4587	4
	soil treatment	47.886	2	<0.001	
	density	52.850	2	<0.001	

## Data Availability

Upon a reasonable request, the authors provided data that supported the findings of this study. This article, along with any supplementary files, includes the entire dataset supporting the findings of this study.

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
