# Peer review of "Effects of Contaminated Soil on the Survival and Growth Performance of European (Populus tremula L.) and Hybrid Aspen (Populus tremula L. × Populus tremuloides Michx.) Clones Based on Stand Density"

_plants, 2022, doi:10.3390/plants11151970_

Round 1
Reviewer 1 Report
The Authors investigated the survival rate of different aspen clones on PAH contaminated soils using different tree densities. They aimed this research as a preliminary investigation for phytoremediation of such soils. Numerous studies investigated the effective application of Populus sp. on soils containing organic contaminants. The manuscript does not highlight the novelty of the present investigation and there are major issues with the applied methods and the presentation of the results.
There is no soil analysis in this work. It is not clear what are properties of the applied soil, what was the concentration of PAH in soil. Hence, there is no information on what the trees react to. If the Authors are able to present this data and evaluate the results in the light of this, the article may be suitable for publication after revision, but should be rejected in its current form.
Line 130: When happened this leak and when were the soils sampled for the experiment?
Line 135: A map of the area should be included in the manuscript. Figure S1 show the experiment design not the area. The bracket opened in this line is not closed.
Line 137: Which soil layer has been sampled? What is the soil type according to WRB?
Line 145: The pH is not presented as mean ±SE.
Line 144-148: Does this mean that the Authors did not analyse the actual soil (contaminated and control) used for this experiment?
Line 147: Adding the percentages of textural fractions the result is above 100%.
Line 152: as control soil not “to control the soil”
Line 154: What is 5 mg/g dose is based on?
Line 173: What were the dimensions of the buckets? What was the size of the soil surface?
Line 181: What happened to the drainage water?
Line 191: Where are the soil data?
Line 204: The Authors define AH but never use it again in the manuscript.
Other questions about the methods:
How were the soils treated before application: grinded, sieved? How was the oil mixed into such a large amount of soil?
What were the climatic conditions of the greenhouse?
Figure 1-5: There are unnecessary lines in the figures that make interpretation difficult. In each figure the a and c diagram should be a column diagram. These a and b diagrams are somehow redundant too. Diagram a shows the average values of diagram c. Why is the sequence of treatments high, low, medium on diagram b? Low, medium, high is more reasonable.
Figure captions: “Error bars show the 95 % confidence interval. Error bar indicates ± standard error (SE). „ Which is true?
Figure 2: Title is height growth, but on the diagram it is only “height”
From these figure it is hard to tell which difference is significant and which is not. Column diagrams indicating the differences with letters would be more appropriate.
Figure 3: Does this show diameter or diameter growth?
Line 331: Is it the weight of a single tree? It seems low.
Figure 4: The unit of mass is g and not gm.
Line 357: Where this 1.81 is coming from?
Line 411: “the soil was less dry” There are no soil data in the manuscript.
Line 477: There are no weather data in the manuscript.
Line 489: I can not see a value of 0.85 in the figure.
Author Response
Reviewer 1:
The Authors investigated the survival rate of different aspen clones on PAH contaminated soils using different tree densities. They aimed this research as a preliminary investigation for phytoremediation of such soils. Numerous studies investigated the effective application of Populus sp. on soils containing organic contaminants. The manuscript does not highlight the novelty of the present investigation and there are major issues with the applied methods and the presentation of the results.
There is no soil analysis in this work. It is not clear what are properties of the applied soil, what was the concentration of PAH in soil. Hence, there is no information on what the trees react to. If the Authors are able to present this data and evaluate the results in the light of this, the article may be suitable for publication after revision, but should be rejected in its current form.
Author’s response:
Many thanks for your valuable comment. We already had a soil analysis of the study site reported in the published articles. In the current version, the information on hydrocarbons concentrations of the study site based on the previous studies is presented in Table 1.
Our main aim of this study is to investigate only the effect of pollutants on plants’ survival rates and growth, not to remove the hydrocarbons through plants; of course, it is valuable in phytoremediation. Contaminated soil treatments reduced the growth and survival rates compared to control, and variations of plant growth and survival rates were visible across soil treatments that could explain how plants reacted against pollutants.
Line 130: When happened this leak and when were the soils sampled for the experiment?
Author’s response:
We reformulated the sentence. Updated sentence:
The creosote preservation facility for railway sleepers was situated in the central part of the study area (~ 1.2 ha) from 1947 to 1958 [37]. During this time, an extensive leak occurred in this area, causing 10,000 liters of creosote to spill into the soil.
In 2013, creosote soil was taken at a depth of 0 to 50 cm from the Somerharju trial square site H14 (Supplementary Figure S1).
Line 135: A map of the area should be included in the manuscript. Figure S1 show the experiment design not the area. The bracket opened in this line is not closed.
Author’s response:
A map of the area is included in the current version of the manuscript (Figure 1)
Line 137: Which soil layer has been sampled? What is the soil type according to WRB?
Author’s response:
We updated the sentence:
In 2013, creosote soil was taken at a depth of 0 to 50 cm from the Somerharju trial square site H14
In Finland, most of the soil is composed of Podzols and Histosols classification of WRB; this is just for your information. To know the basic soi properties are also helpful for plant growth and phytoremediation, as presented in Table 1
Line 145: The pH is not presented as mean ±SE.
Author’s response:
The study area was divided into four blocks (Supplementary Figure S1). In Table 1, we presented the pH of four blocks (mean and SE).
Line 144-148: Does this mean that the Authors did not analyse the actual soil (contaminated and control) used for this experiment?
Author’s response:
Yes, we didn’t do soil analysis as we had the previous soil analysis of the contaminated soil of the study area presented in Table1.
Line 147: Adding the percentages of textural fractions the result is above 100%.
Author’s response:
We fixed the typing mistake, and we apologize for that kind of mistake. Thanks for your valuable comment. The basic soil properties are presented in Table 1.
Line 152: as control soil not “to control the soil”
Author’s response:
Updated as control soil. Sorry for the mistake.
Line 154: What is 5 mg/g dose is based on?
Author’s response:
We updated the sentence.
Finally, the pristine soil was spiked with fresh diesel oil (collected from a local Läyliäinen Neste station) by injecting 50 mL of diesel oil with a syringe into the substrate of each bucket to serve as diesel contaminated soil.
Line 173: What were the dimensions of the buckets? What was the size of the soil surface?
Author’s response:
We reformulated the sentence:
A total of 486 trees were transplanted at three planting densities: one plant per pot (coded as low density), two plants per pot (coded as medium density), and six plants per pot (coded as high density) in 162 10 L plastic buckets (height: 24 cm; top surface diameter: 26 cm; bottom surface diameter: 21 cm), which were filled with ~7–8 kg of soil consisting either of control soil or contaminated soil.
Line 181: What happened to the drainage water?
Author’s response:
Nothing happened. The bottom hole of the bucket facilitated the ensure the proper drainage system when the plants were irrigated with water or rain. Otherwise, water would be deposited in the bucket, and excessive water is not suitable for plant health and even threat to plant death, e.g., root system damage and insect attack.
Line 191: Where are the soil data?
Author’s response:
These two sentences were deleted.
Line 204: The Authors define AH but never use it again in the manuscript.
Author’s response:
AH deleted
Other questions about the methods:
How were the soils treated before application: grinded, sieved? How was the oil mixed into such a large amount of soil?
Author’s response:
Sampled soils have not been sorted, ground, or sieved.
What were the climatic conditions of the greenhouse?
During wintertime, the plants were grown in the model growth condition. In the summertime, plants were exposed to the outside (open greenhouse). We updated the sentences.
Updated sentences are:
The temperature in the storage (model growth condition) was maintained at 22–24 °C with a photoperiod of 16 h light at 20 °C and 8 h dark at 19 °C, and relative humidity at around 80 %.
Climate condition in the open greenhouse:
During the growing season, the mean monthly temperature in the study area ranges from 21 to 25 °C in the summer to – 10 to 22 °C in the winter. The length of daylight varies from 5 hours in winter to 20 hours in summer.
Figure 1-5: There are unnecessary lines in the figures that make interpretation difficult. In each figure, the a and c diagram should be a column diagram. These a and b diagrams are somehow redundant too. Diagram a shows the average values of diagram c. Why is the sequence of treatments high, low, medium on diagram b? Low, medium, high is more reasonable.
Author’s response:
We updated all Figs. It was a problem with Adobe software. We are incredibly sorry for that. Now we have two columns diagram based on your suggestion. As one of the other reviewers also suggested presenting two graphics. In the current version of the manuscript, we have a sequence of treatments with low, medium, and high density.
Figure captions: “Error bars show the 95 % confidence interval. Error bar indicates ± standard error (SE). „ Which is true?
Author’s response:
Error bar indicates ± standard error (SE) is true, and we updated that in the manuscript.
Figure 2: Title is height growth, but on the diagram it is only “height”
Author’s response:
The “ growth” was deleted.
We updated the sentence:
Figure 3. Average (n =3) absolute height (cm) of European aspen and hybrid aspen clones grown in different soil treatments (a) and plant densities (b). Error bar indicates ± standard error (SE).
From these figure it is hard to tell which difference is significant and which is not. Column diagrams indicating the differences with letters would be more appropriate.
Author’s response:
We updated the graphics and presented them as bar graphs based on your suggestion. A pairwise significant comparison was presented in supplementary Table 2.
Figure 3: Does this show diameter or diameter growth?
Author’s response:
We updated the sentence:
Figure 4. Average (n =3) diameter (cm) of European aspen and hybrid aspen clones grown in different soil treatments (a) and plant densities (b). Error bar indicates ± standard error (SE). It will be the only diameter.
Line 331: Is it the weight of a single tree? It seems low.
Author’s response:
No, only stem dry biomass was calculated per pot that was already presented in the methods and materials section. Plants were grown in the stress condition which could be the reason for low biomass production.
Figure 4: The unit of mass is g and not gm.
Author’s response:
We reformulated the sentence:
Figure 5. Average (n =3) dry biomass of stem (g) of European aspen and hybrid aspen clones grown in different soil treatments (a) and plant densities (b). Error bar indicates ± standard error (SE).
Line 357: Where this 1.81 is coming from?
Author’s response:
We deleted this sentence.
Line 411: “the soil was less dry” There are no soil data in the manuscript.
Author’s response:
It was just an outlook observation.
Line 477: There are no weather data in the manuscript.
Author’s response:
We included climate data in the current version of the manuscript. Updated sentences are:
During the growing season, the mean monthly temperature in the study area ranges from 21 to 25 °C in the summer to – 10 to 22 °C in the winter. The length of daylight varies from 5 hours in winter to 20 hours in summer.
The temperature in the storage (model growth condition) was maintained at 22–24 °C with a photoperiod of 16 h light at 20 °C and 8 h dark at 19 °C, and relative humidity at around 80 %.
Line 489: I can not see a value of 0.85 in the figure.
Author’s response:
We updated the sentence:
In this study, all the clones exhibited good photosynthesis efficiency since the Fv/Fm values ranged from 0.75 to 0.78 across all treatments.

Reviewer 2 Report
This paper addresses a topical issue - the phytoremediation of soils contaminated with Polycyclic aromatic hydrocarbons (PAH).
The experiment is extensive, but it has its limitations when it comes to selecting some species of woody plants for phytoremediation. As the authors state in the Conclusions, it is a preliminary test, the results obtained being confirmed only after a greater number of years of investigations.
The presentation of the paper could be more concise, it is not really easy to read.
Regarding the height of the poplar individuals resulting from the experiment, as a personal opinion - I think the absolute height seems low, considering that in the literature the growth rate for Hybrid Aspen Clones (Populus tremula × Populus tremuloides) is documented as between 4 and 6 cm / week. What is the total height of the poplar individuals grown in pots after 2 years? What is the correlation between the actual height of individuals and the Absolute Height (AH)?
In the Conclusions chapter I consider that the authors should refer to their own study (listed at No. 37 in References), where the same clones are used in a field study (also polluted with hydrocarbons, from the same source (?) - composition was described by by Mukherjee et al., 2014 in both papers). This is all the more necessary as the results are similar in terms of survival (hybrid aspen clone 291 and European aspen clone R3).
Given that the experiment ended in 2015 and you found the need for future studies - have they been done in these 7 years?
rows 192-193 "the removal percentage of hydrocarbons" - did you estimate the degree of removal of hydrocarbons from the soil during the experiment? Figure S1 in the Supplementary material shows the layout of experimental design.
Below you will find some small formal remarks:
row 83 - ”proliferation (up to 15–50 m)” - please specify if 15-50 m means the height of the tree.
rows 166-167 - please check if the use of the term "seedlings" is appropriate; however, these are plants that are almost 1 year old. Did you have poplar specimens 0.5 cm tall after a year of growing? Some photos taken during the experiment would be useful for paper.
rows 159-167 - please specify in the chapter 2.2 "Plant material" the source of the plant material - the two European aspen and four hybrid aspen clones.
rows 171, 178 - please use constantly ”Four hundred eighty-six” or ”486”.
rows 300-301 - ”Hybrid aspen clone 14 displayed higher diameter growth than other hybrid aspen clones by 33‒ 44 %” - repetitive, please reformulate.
Author Response
This paper addresses a topical issue - the phytoremediation of soils contaminated with Polycyclic aromatic hydrocarbons (PAH).
The experiment is extensive, but it has its limitations when it comes to selecting some species of woody plants for phytoremediation. As the authors state in the Conclusions, it is a preliminary test, the results obtained being confirmed only after a greater number of years of investigations.
The presentation of the paper could be more concise, it is not really easy to read.
Regarding the height of the poplar individuals resulting from the experiment, as a personal opinion - I think the absolute height seems low, considering that in the literature the growth rate for Hybrid Aspen Clones (Populus tremula × Populus tremuloides) is documented as between 4 and 6 cm / week. What is the total height of the poplar individuals grown in pots after 2 years? What is the correlation between the actual height of individuals and the Absolute Height (AH)?
Author’s response:
The plants were grown under stress conditions, for example, pollutant toxicity, and unfavorable growth factors, this might reduce the growth. The correlation is positive. More details are in Fig below:
In the Conclusions chapter I consider that the authors should refer to their own study (listed at No. 37 in References), where the same clones are used in a field study (also polluted with hydrocarbons, from the same source (?) - composition was described by by Mukherjee et al., 2014 in both papers). This is all the more necessary as the results are similar in terms of survival (hybrid aspen clone 291 and European aspen clone R3).
Author’s response:
In the current version of the manuscript, we cited the reference [37] in the conclusion section on the survival rates and growth based on your suggestion.
Yes, the same clones and soil materials were used in both greenhouse and field experiments. Thanks for your valuable comment.
Given that the experiment ended in 2015 and you found the need for future studies - have they been done in these 7 years?
Author’s response:
Field experiments based on the same clones and same polluted soil are ongoing since 2011. Lack of funding is also an issue to start another experiment.
rows 192-193 "the removal percentage of hydrocarbons" - did you estimate the degree of removal of hydrocarbons from the soil during the experiment? Figure S1 in the Supplementary material shows the layout of experimental design.
Author’s response:
We deleted these sentences in the current version of the manuscript. Thanks for your valuable comment.
Below you will find some small formal remarks:
row 83 - ”proliferation (up to 15–50 m)” - please specify if 15-50 m means the height of the tree.
Author’s response:
We updated the sentence. Reformulated sentence:
European aspen and hybrid aspen have some unique characteristics that are suitable for phytoremediation experiments, such as quick physiological responses to environmental factors, ease of establishment, fast-growing with a height of 15-50 m, and deep root systems [23, 26, 29, 30].
rows 166-167 - please check if the use of the term "seedlings" is appropriate; however, these are plants that are almost 1 year old. Did you have poplar specimens 0.5 cm tall after a year of growing? Some photos taken during the experiment would be useful for paper.
Author’s response:
Unfortunately, we don’t have poplar specimens 0.5 cm tall after a year of growth. One photo taken during the experiment is presented in Fig. S3 (supplementary material).
rows 159-167 - please specify in the chapter 2.2 "Plant material" the source of the plant material - the two European aspen and four hybrid aspen clones.
Author’s response:
We reformulated the sentences. Reformulated sentences are:
We used two European aspen and four hybrid aspen clone seedlings. In August 2012, these seedlings were produced using the micropropagation method [51] in the Haapastensyrjä unit laboratory. Parent plants were collected from several aspen trials in southern Finland. Thanks for your valuable comment.
rows 171, 178 - please use constantly ”Four hundred eighty-six” or ”486”.
Author’s response:
At the beginning of the sentence, we spelled out otherwise not. Now we updated the sentence to confusion. Updated sentence:
Altogether 486 were obtained from two European aspen clones (coded as R3, R4) and four hybrid aspen clones (coded as 14, 27, 191, 291).
rows 300-301 - ”Hybrid aspen clone 14 displayed higher diameter growth than other hybrid aspen clones by 33‒ 44 %” - repetitive, please reformulate.
Author’s response:
We deleted this sentence in the current version of the manuscript.

Reviewer 3 Report
Dear Sirs,
I found your manuscript well structured as regards experimental design and variables investigated, but, as I wrote in comments attached in the pdf file, I think it lacks assessments about the translocation of pollutants or derived compounds in stems or leaves, i.e. the phytoextraction topic. Thus, altough its scientific soundness is adequate, I cannot say the same for its overall merit. Anyway, I think it may be interesting for publication as propaedeutic for deeper investigations. The quality of figures can be improved, considering that lines and error bars are often confused and disturbed by unessential graphical elements. You can find other minor remarks in the attached manuscript pdf file.
I recommend your manuscript for publication after minor revision, best regards.

Author Response
Reviewer 3:
Dear Sirs,
I found your manuscript well structured as regards experimental design and variables investigated, but, as I wrote in comments attached in the pdf file, I think it lacks assessments about the translocation of pollutants or derived compounds in stems or leaves, i.e. the phytoextraction topic. Thus, altough its scientific soundness is adequate, I cannot say the same for its overall merit. Anyway, I think it may be interesting for publication as propaedeutic for deeper investigations. The quality of figures can be improved, considering that lines and error bars are often confused and disturbed by unessential graphical elements. You can find other minor remarks in the attached manuscript pdf file.
I recommend your manuscript for publication after minor revision, best regards.
Lines 20-21 - Sentence not clear
Author’s response:
We updated the sentence. Reformulated sentence:
Low plant density resulted in a higher survival rate and growth than the other two density treatments. In contrast, these two density treatments did not affect the plant survival rate and growth to a greater extent.
Line 23 - This clone doesn't appear in those cited in lines 11 and 12
Author’s response:
We apologize for that kind of mistake. Thank you for your valuable comment. Updated sentence (lines 11-12):
This study was conducted to assess the survival rates, growth, and chlorophyll fluorescence (Fv/Fm) of four hybrid aspen (14, 191, 27, 291) and two European aspen (R3 and R4) clones cultivated in creosote- and diesel oil-contaminated soil treatments under three different plant densities: one plant per pot (low density), two plants per pot (medium density), and six plants per pot (high density) over a 2-year three months period.
Line 75 - Correct sentence: "Belong to Salicaceae family. By their scientific name, it is obvious they are Populus.
Author’s response:
We corrected the sentence. Thank you for your valuable comment. Corrected sentence: European aspen and hybrid aspen belong to the Salicaceae family.
Line 133 - Microgram symbol must be without dot
Author’s response:
Microgram symbol presented without dot based on your suggestion in the current version of the manuscript. Thanks for your valuable comment.
Line 189 - In Fig. S1 you indicate the highest density as "d6", whereas in its caption you indicate it as "d3". Uniform the symbols
Author’s response:
In the caption, d3 is replaced by d6 in the current version of the manuscript.
Line 220 – Repetition
Author’s response:
A repetitive part was deleted in the current version of the manuscript. thank you for your valuable comment.
Line 223 - Citate correctly: for example, "R Statistical Software (v4.1.2; R Core Team 2021)"
Author’s response:
Your recommended citation style is used in the current version of the manuscript. Thanks for your valuable comment.
Line 515-517 - Another limitation is that nothing was investigated about the phytoextractive power, i.e. the accumulation of pollutants in stems or leaves. In this case pollutants are organic and probably are partially absorbed in soil or modified by plant enzymes or metabolized, but maybe some heavy metals present in PAH are translocated to tissues. It would be advisable to assess and quantify such translocation
Author’s response:
We agree with your valuable comment. In this study, our main aim was the only evaluation of plant survival and growth under hydrocarbon-polluted soil.

Author Response
Reviewer 4:
See the attached PDF file.
Reviewer-4
The paper presents the results of a major pot experiment over more than two years. Growth parameters and chlorophyll fluorescence were monitored in three different planting densities of four hybrid and two European aspen clones in two types of polluted soil and clean control soil. The paper addresses many variables and presents a lot of data, but the text should be elaborated and the graphs should be improved. The results on planting density are interesting and important for further development of phytoremediation technologies; however, the soil analyses of the targeted pollutants are lacking.
Author’s response :
Thanks for your valuable comments. Soil analysis has already been done in previous studies based on the field trial and results published, for instance, Salam et al. 2020. Our main aim of this study is only to see the effect of density and hydrocarbon concentration on the survival, growth, and Fv/Fm values of aspen clones. Those are important parameters for phytoremediation experiments. However, the hydrocarbons (HC) concentrations are presented in Table 1 based on the previous studies in the current version of the manuscript. We agree that soil analysis after the phytoremediation experiment would be beneficial to evaluate the reduction of HC concentrations from the soil through plants.
First, the language should be improved, especially with regard to sentence structure and what refers to what. Also, there is a lot of repetition of information and double spaces in the paper. Many sentences could be optimized.
Author’s response:
We updated the manuscript based on the reviewer’s comment. Thanks for your valuable comment.
Second, all the figures in the MS need to be improved. There are weird diagonal lines in all the graphs that must be removed. I do not see the point of presenting average results (for all treatments together) when individual treatments are already presented.
Author’s response:
We apologized for that kind of mistake. It was a mistake of Adobe software. In the current version, we checked carefully to avoid such kinds of mistakes. We improved the graphics and now we presented based on the reviewer comments. Thanks for your valuable comment.
This would allow two graphs per parameter and a more concise presentation of the results. The data should also be presented as scattered graphs with no lines between data sets.
Author’s response:
In the current version, we have only two graphs instead of three. Now we presented a column bar graph as one of the other reviewers suggested. Thanks for your valuable comment.
The graphs representing different densities should have an order of increasing density on the x-axis (low, medium, high and not high, low, medium).
Author’s response:
We updated the current version of the manuscript based on the reviewer’s comment having an order of low, medium, and high in graphs and in the text.
Third, the discussion repeats the results and other information too often.
Author’s response:
We updated the manuscript based on the reviewer’s comments. Careful has been taken to avoid repetition. Thanks for your valuable comment.
In the conclusion, you claim that there was no budget for pollutant analyses. This should be addressed in the discussion (part about phytoremediation). Conclusions should highlight the main findings.
Author’s response:
This whole part moved into the discussion section and highlighted the main findings in the conclusions section. Thanks for your valuable comment.
Line 12: correct the clones of hybrid aspen – instead of R4 it's probably 14
Author’s response:
Updated.
Line 18: add survival rate for diesel-contaminated soil
Author’s response:
Added this information to the manuscript. Updated sentences are:
The results revealed that contaminated soils affected all plants' survival rates and growth. However, plants grown in the creosote-contaminated soil displayed a 99 % survival rate, whereas plants cultivated in the diesel-contaminated soil showed a 22–59 % survival rate.
Lines 20-21: What are »these two factors«?
Author’s response:
We are sorry for making confusion. We updated the text now. These two factors are replaced by these two density treatments. Updated text:
In contrast, these two density treatments did not affect the plant survival rate and growth to a greater extent, particularly in contaminated soil treatments.
Line 84: »the best species among other fast-growing species« - according to what? Phytoremediation or biomass production or…?
Author’s response:
We deleted the confusing part. Updated sentence is
European aspen and hybrid aspen have some unique characteristics that are suitable for phytoremediation experiments, such as quick physiological responses to environmental factors, ease of establishment, fast-growing with a height of 15-50 m, and deep root systems [23, 26, 29, 30].
Line 132: How is elevated PAH deposit 10-11 m beneath the groundwater? Maybe in the groundwater 10-11 m below the surface?
Author’s response:
We updated the sentence:
The elevated PAH and TPH deposits near the groundwater are approximately 10–11 m below the soil surface.
Thanks for your valuable comment.
Line 133: Change »Until« with »Up to«
Author’s response:
Changed until with up to
Line 135: reference to the figure is not clear and the full bracket is missing
Author’s response:
Now Fig. is presented as a Supplementary Figure (Fig. S1) in the current version.
Line 139: The 16 priority pollutants are probably the 15 listed PAHs and another parameter is TPH? Full brackets are also missing
Author’s response:
We reformulated the sentence
(total PAHs, 16 USEPA priority PAHs, TPH (C10–C21, C22–C40, and C10–C40)),
TPH: C10–C21, C10–C40, and C22–C40
For more details about PAHs and TPH, please see:
Kuppusamy et al. 2020; 10.1007/978-3-030-24035-6_1
Salam et al. 2020, reference: [37]
Mukherjee et al. 2014, reference: [50]
Lines 145-147: I suggest putting soil properties in a table
Author’s response:
Soil properties are presented in Table 1 in the current version of the manuscript.
Line 157 (Figure 1): Figure should appear later in the text in the Results section.
Author’s response:
The figure moved to the results section.
Line 137: There were 486 trees in three different densities – how you then have 486 buckets?
Author’s response:
We are extremely sorry for such kind of mistake. It would be 162 buckets. We updated the text. Thanks for your valuable comment.
Line 178: Write the number with numbers not in text
Author’s response:
We agree with your comment. However, beginning of the sentence we spelled out otherwise we used numbers.
Line 241: change »Hybrid« with »hybrid« and »29l« with »291«
Author’s response:
changed »Hybrid« with »hybrid« and »29l« with »291«
Line 244-246: by not checking the graph this sentence is not clear to the reader
Author’s response:
We deleted the sentences. Reformulated sentences:
The plants cultivated in low-density (one plant per pot) and medium-density (two plants per pot) treatments had a 10 % higher survival rate than those cultivated in high-density treatment (six plants per pot) (Figure 2b).
Line 259: additional caption to Table 1 is not needed. Delete »Table 1«
Author’s response:
Deleted Table 1
Line 262 (Table 1): put factors in the same order for each variable
Author’s response:
We updated based on the reviewer’s comment.
Table 2. Analysis of deviance (ANOVA)
|
variable |
factors |
χ2 |
DF (degree of freedom) |
p-value |
model |
|
survival rate |
clone |
l2.3 |
5 |
0.0312 |
1 |
|
soil treatment |
93.2 |
2 |
< 0.001 |
||
|
density |
7.2 |
2 |
0.0272 |
||
|
absolute height |
clone |
106.99 |
5 |
< 0.001 |
2 |
|
soil treatment |
15.05 |
2 |
< 0.001 |
||
|
density |
35.65 |
2 |
< 0.001 |
||
|
diameter |
423.22 |
1 |
< 0.001 |
||
|
diameter |
clone |
250.68 |
5 |
< 0.001 |
3 |
|
soil treatment |
20.88 |
2 |
< 0.001 |
||
|
density |
1.43 |
2 |
0.4893 |
||
|
height |
425.7 |
1 |
< 0.001 |
||
|
stem dry biomass |
clone |
4.660 |
5 |
0.4587 |
4 |
|
soil treatment |
47.886 |
2 |
< 0.001 |
||
|
|
density |
52.850 |
2 |
< 0.001 |
|
Line 246: Reformulate the sentence – hybrid aspen clones were 5-40% higher than European…
Line 266: Reformulate the sentence – clone R3 was only 6% higher than R4?
Line 267: »Higher height growth was observed…« - delete height
Line 270: What is more significant height growth? Plants in low-density treatment were significantly higher than plants in high- and medium-density treatments?
Lines 287-292: it's difficult to read this part. I suggest reporting a range and make sentences clearer.
Lines 320-324: it's difficult to read this part. I suggest reporting a range.
Author’s response:
We reformulated the whole paragraph. Updated paragraph:
Contaminated soil treatments reduced the growth compared to control in all clones (Figure 3a). The diesel-contaminated soil treatment decreased absolute height by 5–33 % for hybrid aspen clones (14, 191, 27, and 291) and 28–44 % for European aspen clones (R3 and R4) when compared to their respective controls. Creosote-contaminated soil treatment also decreased absolute height by 19–38 % for hybrid aspen clones and 20–22 % for European aspen clones compared to their respective controls (Figure 3a). Superior growth was apparent in clone 14 compared to other hybrid aspen clones grown in all soil treatments, whereas clone R3 was superior to R4 in growth when plants were cultivated in creosote-contaminated and controlled soil treatments. In contrast, clone R4 was superior to R3 in terms of growth when plants were grown in diesel-contaminated soil treatment (Figure 3a). A decreasing trend in the growth of the plants was evident from the low-density treatment to high-density treatments (Figure 3b). Significantly 19–33 % higher growth was observed in the plants cultivated in low-density treatment than plants cultivated in medium- and high-density treatments.
Line 271-279: report in the order of low, medium and high-density treatments
Author’s response:
In the current version, we have followed the order of low, medium, and high-density treatments
Lines 335-336: are the grams reported a total weight of one plant or of total biomass in the pot? Report the numbers for medium and heigh at the end of the sentence and add »respectively«
Author’s response:
The g reported the total biomass of the stem in the pot. We reformulated the sentence.
Reformulated sentence:
Plants in low-density treatment exhibited the highest dry biomass, followed by medium-density and high-density treatments (4.59, 2.12, and 1.32 g pot‒1), respectively.
Line 349 (Figure 4): Why the density is not reported by treatments like for other parameters? What unit is »gm«? Grams? Abbreviation for grams is »g«
Author’s response:
In the current version of the manuscript, we presented it as “g”
Line 392: Change »29l« with »291«
Author’s response:
Replaced.
Line 409-410: maximum dry density and optimum water content in the soil? Please refer to the study where soil data is reported. »There was also reduction in permeability and strength« - of what?
Author’s response:
We reformulated the sentence:
Increasing the crude oil content reduced the maximum dry density and optimum water content in the soil. There was also a reduction in permeability and strength of the soil [57].
Line 431-432: the listed parameters do not seem to be linked only to hydrocarbon-contaminated soils but also to other site-specific parameters. Please check the cited references.
Author’s response:
We updated with a new reference.
Haider, F.U.; Ejaz, M.; Cheema, S.A.; Khan, M.I.; Zhao, B.; Liqun, C.; Salim, M.A.; Naveed, M.; Khan, N.; Núñez-Delgado, A.; Mustafa, A. Phytotoxicity of petroleum hydrocarbons: Sources, impacts and remediation strategies. Environmental Research. 2021, 197, 111031.
Line 436: the first sentence is unfinished
Author’s response:
We reformulated the sentence. Updated sentence:
To achieve consistently high remediation efficiency, planting density should be considered.
Line 435: reference is missing – which field studies?
Author’s response:
We reformulated the sentence:
In phytoremediation studies, Nguemté et al. [62] and Shirdam et al. [63] reported that toxic compounds can inhibit plant growth in the presence of total petroleum hydrocarbons.

Round 2
Reviewer 1 Report
The Authors have significantly improved the manuscript. However, some questions are still open. I still have concerns about whether the article is suitable for publication without knowledge of the contaminants (soil data are only known from a previous study – according to table 1 the differences between min and max values can be significant; how does fresh oil contamination campare to the old one?).
1) Previous comment: What is 5 mg/g dose is based on?
This question sought to suggest that a theoretical background should be given to this value. Why was this dose chosen? How do the contaminants in it relate to the concentrations of the substances measured in the already contaminated soil? It would be good to see this so that we know what we are comparing to what.
Author’s response: “Finally, the pristine soil was spiked with fresh diesel oil (collected from a local Läyliäinen Neste station) by injecting 50 mL of diesel oil with a syringe into the substrate of each bucket to serve as diesel contaminated soil.” Does this mean that the oil was not mixed into the soil? In this way how can you assure that the contamination is evenly distributed in the soil?
2) Previous comment: From these figure it is hard to tell which difference is significant and which is not. Column diagrams indicating the differences with letters would be more appropriate.
Author’s response: “We updated the graphics and presented them as bar graphs based on your suggestion. A pairwise significant comparison was presented in supplementary Table 2. “
It is supplementary table 1. I still suggest to indicate significant differences on the graphs not in a supplementary file since it is important for the evaluation of the results.
3) Previous comment: “the soil was less dry” There are no soil data in the manuscript.
Author’s response: "It was just an outlook observation."
The evaluation of the experiment should be based on measured data. This way for example one can say without measurement that the tress were higher or thicker in this treatment than in the other. But this method is at least questionable. I think that these sentences should be deleted.
4) Now there is a picture about the trees in the supplementum from the last year of the experiment. The trees are very tiny compared to that their reported diameter in the manuscript is 3-6 cm. Could it not be more the circumference?
Line 411: Correct “….”
Author Response
The Authors have significantly improved the manuscript. However, some questions are still open. I still have concerns about whether the article is suitable for publication without knowledge of the contaminants (soil data are only known from a previous study – according to table 1 the differences between min and max values can be significant; how does fresh oil contamination campare to the old one?).
1) Previous comment: What is 5 mg/g dose is based on?
This question sought to suggest that a theoretical background should be given to this value. Why was this dose chosen? How do the contaminants in it relate to the concentrations of the substances measured in the already contaminated soil? It would be good to see this so that we know what we are comparing to what.
Author’s response: “Finally, the pristine soil was spiked with fresh diesel oil (collected from a local Läyliäinen Neste station) by injecting 50 mL of diesel oil with a syringe into the substrate of each bucket to serve as diesel contaminated soil.” Does this mean that the oil was not mixed into the soil? In this way how can you assure that the contamination is evenly distributed in the soil?
Author’s response:
We updated the sentence:
Finally, the pristine soil was spiked with fresh diesel oil (collected from a local Läyliäinen Neste station) at the rate of 5 mg g-1 DW soil by injecting a 50 mL syringe in the concrete mixer to serve as diesel-contaminated soil. Approximately 30-40 kg of dry soil were used in the concrete mixer per round, and the duration of mixing was 2-3 minutes per round to combine dry soil with fresh diesel oil.
This dose (5 mg g-1 DW soil) results in a normal concentration that can commonly be found in contaminated road sites in Finland. This dose is also previously used in a university experiment (results unpublished). However, fresh diesel at minimal doses is toxic to plants. Previous results didn’t publish yet, therefore, no background value is available for fresh diesel oil.
2) Previous comment: From these figure it is hard to tell which difference is significant and which is not. Column diagrams indicating the differences with letters would be more appropriate.
Author’s response: “We updated the graphics and presented them as bar graphs based on your suggestion. A pairwise significant comparison was presented in supplementary Table 2. “
It is supplementary table 1. I still suggest to indicate significant differences on the graphs not in a supplementary file since it is important for the evaluation of the results.
Author’s response:
We reanalyzed and indicated significant differences in the graphs
Means within clones followed by the same lower-case letters are not significantly different (p > 0.05). Means between density treatments followed by the same lower-case letters are also not significantly different (p > 0.05).
3) Previous comment: “the soil was less dry” There are no soil data in the manuscript.
Author’s response: "It was just an outlook observation."
The evaluation of the experiment should be based on measured data. This way for example one can say without measurement that the tress were higher or thicker in this treatment than in the other. But this method is at least questionable. I think that these sentences should be deleted.
Author’s response:
These sentences were deleted.
4) Now there is a picture about the trees in the supplementum from the last year of the experiment. The trees are very tiny compared to that their reported diameter in the manuscript is 3-6 cm. Could it not be more the circumference?
Author’s response:
The image is not presented by the actual size of an image in the manuscript. Without accurate image details, it may not be possible to get the real size of the diameter. This image is presented to see the greenhouse experiment, not to calculate the size of the diameter.
Line 411: Correct “….”
Author’s response:
Corrected. Extra dots were deleted

Reviewer 4 Report
The paper was improved compared to the first review round. Additional changes should be applied as follows.
L21: substitute »these two density« with »the medium and high density«
L46: Substitute »Polycyclic aromatic hydrocarbons« with PAH as the abbreviation was already defined.
L94-95: The two sentences are not clearly written. Do they refer to the same study?
L134: Which months are referred to as »summer« and which as »winter«?
L164: Here it is not yet known that the plants were in buckets, please rephrase the sentence. Add also the volume of the soil to which 50 mL of diesel was added.
L179-183: The described timeline is not clear. »The plants were stored inside during winter period« - which one – in 2012, 2013? When exactly is the winter period (start and end month)?
L189: Use either »pot« or »bucket« to avoid confusion
L197: 486 trees
L271: Titles of the axes should start with a capital letter. Add (n) fort he b) part of the graph.
The comments go to all graphs.
L323-324: Improve the sentence grammatically.
L347: …produced significantly more (53-71%) biomass than…
L373: How would you explain the opposite trend of Fv/Fm ratio (increasing from low to high density) compared to the trends in growth and biomass production (decreasing from low to high density). Add additional explanation to L509.
L396: Delete (HC) as the abbreviation is not used later on.
L411: Delete the excess dots.
L460: Delete the excess dots.
Author Response
Comments and Suggestions for Authors
The paper was improved compared to the first review round. Additional changes should be applied as follows.
L21: substitute »these two density« with »the medium and high density«
Author’s response:
Thanks for your valuable comments. Updated the sentence.
In contrast, the medium- and high-density treatments did not affect the plant survival rate and growth to a greater extent, particularly in contaminated soil treatments.
L46: Substitute »Polycyclic aromatic hydrocarbons« with PAH as the abbreviation was already defined.
Author’s response:
Substituted with PAH
The PAH and total petroleum hydrocarbons (TPH) contaminants pollute surface water, groundwater, soil, and sediments, threatening the ecosystem and urgently need to be solved [8].
L94-95: The two sentences are not clearly written. Do they refer to the same study?
Author’s response:
They are not the same study. We updated the sentence:
In another phytoremediation study, Populus plants were used for leachate and wastewater treatments, and they generally showed better growth and resistance than the Salix plants [36].
L134: Which months are referred to as »summer« and which as »winter«?
Author’s response:
We updated the sentence:
During the growing season, the mean monthly temperature in the study area ranges from 21 to 25 °C in the summer (May-August) and from –10 to 22 °C in the winter (November-March).
L164: Here it is not yet known that the plants were in buckets, please rephrase the sentence. Add also the volume of the soil to which 50 mL of diesel was added.
Author’s response:
Finally, the pristine soil was spiked with fresh diesel oil (collected from a local Läyliäinen Neste station) at the rate of 5 mg g-1 DW soil by injecting a 50 mL syringe in the concrete mixer to serve as diesel-contaminated soil. Approximately 30-40 kg of dry soil were used in the concrete mixer per round, and the duration of mixing was 2-3 minutes per round to combine dry soil with fresh diesel oil.
L179-183: The described timeline is not clear. »The plants were stored inside during winter period« - which one – in 2012, 2013? When exactly is the winter period (start and end month)?
Author’s response:
We updated the paragraph:
The seedlings were transplanted into boxes at room temperature in February 2013 and kept in the boxes until June 2013. There were no signs of biological activity, i.e., no more green leaves appeared. Finally, seedlings were planted in pots for the greenhouse experiment, and their height ranged from approximately 0.5 cm to l5 cm depending on the clone.
L189: Use either »pot« or »bucket« to avoid confusion
Author’s response:
We used only pot throughout the manuscript to avoid confusion
L197: 486 trees
Author’s response:
Added trees
L271: Titles of the axes should start with a capital letter. Add (n) fort he b) part of the graph.
The comments go to all graphs.
L323-324: Improve the sentence grammatically.
Author’s response:
Updated the sentence:
As compared with clone 191 grown in control soil, its diameter declined by 20 % in contaminated soil (Figure 4a).
L347: …produced significantly more (53-71%) biomass than…
Author’s response:
Updated the sentence based on your suggestion.
produced significantly more (53-71%) biomass than
L373: How would you explain the opposite trend of Fv/Fm ratio (increasing from low to high density) compared to the trends in growth and biomass production (decreasing from low to high density). Add additional explanation to L509.
Author’s response:
A lower Fv/Fm value was found in plants in the low-density treatment in comparison to plants in the medium- and high-density treatments (Fig. 6b). This could be explained by the low-stress conditions experienced by plants in the low-density treatment, or perhaps plants didn't need to fully utilize photosynthetic activities for growth [7].
L396: Delete (HC) as the abbreviation is not used later on.
Author’s response:
Deleted HC
L411: Delete the excess dots.
Author’s response:
Deleted
L460: Delete the excess dots.
Author’s response:
Deleted

Round 3
Reviewer 1 Report
The Authors improved the figures with the results of statistical analysis. However, in some cases the text explaining the results of statistics are not coherent with the applied analysis. The Authors made comparisons within clones and not between clones. But this can not be seen in the text.
Comparisons like "Superior growth was apparent in clone 14 compared to other hybrid aspen clones grown in all soil treatments, whereas clone R3 was superior to R4 in growth when plants were cultivated in creosote-contaminated and controlled soil treatments." are not supported by the figures since the differences are shown only within clones: "Means within clones followed by the same lower-case letters are not significantly different" The text should be revised and reworded according to the applied statistics.
Citations in conclusions should be omitted.
Author Response
Third revision
Reviewer comment 1:
The Authors improved the figures with the results of statistical analysis. However, in some cases the text explaining the results of statistics are not coherent with the applied analysis. The Authors made comparisons within clones and not between clones. But this can not be seen in the text.
Comparisons like "Superior growth was apparent in clone 14 compared to other hybrid aspen clones grown in all soil treatments, whereas clone R3 was superior to R4 in growth when plants were cultivated in creosote-contaminated and controlled soil treatments." are not supported by the figures since the differences are shown only within clones: "Means within clones followed by the same lower-case letters are not significantly different" The text should be revised and reworded according to the applied statistics.
Author’s response: done
Reviewer comment 2:
Citations in conclusions should be omitted.
Author’s response: done